# Artificial Intelligence That Predicts Sensitizing Potential of Cosmetic Ingredients with Accuracy Comparable to Animal and In Vitro Tests—How Does the Infotechnomics Compare to Other “Omics” in the Cosmetics Safety Assessment?

**DOI:** 10.3390/ijms24076801

**Published:** 2023-04-06

**Authors:** Jadwiga Kalicińska, Barbara Wiśniowska, Sebastian Polak, Radoslaw Spiewak

**Affiliations:** 1Department of Experimental Dermatology and Cosmetology, Jagiellonian University Medical College, ul. Medyczna 9, 30-688 Krakow, Poland; 2Department of Social Pharmacy, Jagiellonian University Medical College, ul. Medyczna 9, 30-688 Krakow, Poland

**Keywords:** cosmetic ingredients, sensitizing potential, contact allergy, risk assessment, infotechnomics, in silico modelling

## Abstract

The aim of the current study was to develop an in silico model to predict the sensitizing potential of cosmetic ingredients based on their physicochemical characteristics and to compare the predictions with historical animal data and results from “omics”-based in vitro studies. An in silico model was developed with the use of WEKA machine learning software fed with physicochemical and structural descriptors of haptens and trained with data from published epidemiological studies compiled into estimated odds ratio (eOR) and estimated attributable risk (eAR) indices. The outcome classification was compared to the results of animal studies and in vitro tests. Of all the models tested, the best results were obtained for the Naive Bayes classifier trained with 24 physicochemical descriptors and eAR, which yielded an accuracy of 86%, sensitivity of 80%, and specificity of 90%. This model was subsequently used to predict the sensitizing potential of 15 emerging and less-studied haptens, of which 7 were classified as sensitizers: cyclamen aldehyde, N,N-dimethylacrylamide, dimethylthiocarbamyl benzothiazole sulphide, geraniol hydroperoxide, isobornyl acrylate, neral, and prenyl caffeate. The best-performing model (NaiveBayes eAR, 24 parameters), along with an alternative model based on eOR (Random Comittee eOR, 17 parameters), are available for further tests by interested readers. In conclusion, the proposed infotechnomics approach allows for a prediction of the sensitizing potential of cosmetic ingredients (and possibly also other haptens) with accuracy comparable to historical animal tests and in vitro tests used nowadays. In silico models consume little resources, are free of ethical concerns, and can provide results for multiple chemicals almost instantly; therefore, the proposed approach seems useful in the safety assessment of cosmetics.

## 1. Introduction

Cosmetics are a relevant cause of allergic contact dermatitis (ACD). In a recent study of Brazilian patients with suspected ACD, cosmetics were confirmed as the ultimate cause of the disease in 16.5% patients, of whom 89.7% were women [1]. A multi-center Europe-wide study of 39 cosmetic ingredients showed that the top sensitizers were sodium metabisulfite (sensitization rate 3.98%), lanolin (Amerchol L-101, 3.68%), and propolis (3.26%) [2]. The frequency of allergic reactions to cosmetic ingredients prompts appropriate action. The European Chemicals Agency (ECHA) has introduced the REACH regulation governing the registration, evaluation, authorization, and restriction of chemicals in order to better protect human health and the environment from chemical risks [3]. Regulation (EC) No 1223/2009 of the European Parliament and of the Council of 30 November 2009 states that “a cosmetic product made available on the market shall be safe for human health when used under normal or reasonably foreseeable conditions of use” [4]. Epidemiological data confirm that regulatory restrictions on the use of ingredients with known sensitizing properties can result in a reduction of sensitization rates [5]. This positive effect may be partly hampered by the fact that cosmetic products sometimes contain undeclared (“hidden”) ingredients or contaminants with possible adverse effects going beyond of what would be expected based on their declared content [6,7].

Before 2004, cosmetics and their ingredients had mainly been tested on animals. Animal tests were designed to determine repeated dose toxicity, reproductive and developmental toxicity, mutagenicity and—most importantly for the topic of this study—skin-sensitizing potential. The most popular animal tests to assess the sensitizing potential of cosmetic ingredients were the Bühler test, Guinea Pig Maximization Test (GPMT), and Local Lymph Node Assay (LLNA). In the Bühler test and GPMT, albino guinea pigs were the animals of choice, while mice were used in LLNA test [8]. In Bühler test, animals were exposed to high doses of tested substances applied on shaved skin in three series of 6 h, every 7 days each (one patch per week) [9]. The skin was observed for the occurrence of erythema, edema, or necrosis [8]. The GPMT protocol was divided into two stages: intradermal injection of tested substances with or without Freund’s Complete Adjuvant (FCA), and a topical exposure after 7 days [8,9,10]. The skin reaction was evaluated similarly to the Bühler test. LLNA was based on the application of a tested substance onto the ears of mice for 3 consecutive days. On day 6, the animal’s lymph nodes were retrieved to measure the outcome. The principle of the LLNA was that sensitizers induce the proliferation of lymphocytes in the lymph nodes. This proliferation is proportional to the dose and to the sensitizing potency of the applied chemical, which provides a simple means of measuring sensitization, either by weighing the lymph nodes or, more accurately, measuring their ^3^H-methyl thymidine incorporation. The ratio of the mean proliferation in each treated group divided by proliferation in the vehicle-treated control (VC) group was termed as the Stimulation Index (SI). With SI ≥ 3, a test substance was considered as sensitizing [11]. In practice, to allow for a better comparison of sensitizers, the value of EC3 was adopted, defined as the concentration of a tested substance required to elicit a three-fold increase in lymph node proliferation as compared to control animals [12]. If the EC3 value was ≤0.2%, the test substance was recognized as an extreme sensitizer; between 0.2% and 2%, as a strong sensitizer; and EC3 > 2% was characterized a moderate sensitizer [12,13].

The European Council’s Directive 2003/15/EC of 27 February 2003 has imposed an ultimate end on the testing of cosmetics and their ingredients on animals. After consulting with the Scientific Committee on Consumer Safety (SCCS) and the European Centre for the Validation of Alternative Methods (ECVAM), the European Commission has set a schedule to phase out animal testing. On 1 October 2004, a complete ban came into effect on the animal testing of finished cosmetic products and their ingredients, provided there was an approved and adequate alternative method. On 11 March 2009, an unconditional ban on all tests of cosmetics ingredients on animals was introduced in the EU with the exception of toxicokinetics, repeated dose toxicity, and reproductive toxicity studies. Finally, on 11 March 2013, a total ban on testing cosmetics ingredients and finished cosmetics in animals has come into force, along with a ban on the placing products tested on animals on the market, regardless of the availability of alternative methods (Table 1) [5]. 

Alternative in vitro methods designed to replace animal experiments in the evaluation of toxicity and allergenicity of cosmetic ingredients are based on the “omics” approach with the goal of assessing the key events that warn of possible risks at the molecular, structural, or cellular stages (key events 1–3) in the adverse outcome pathway, rather than at the organ level (e.g., swelling of lymph nodes, key event 4) and skin symptoms (adverse outcome) as was used in the case of animal tests [14]. The test systems nowadays are typically homogeneous cultures of cells derived from human skin—melanocytes, fibroblasts, keratinocytes, Langerhans cells, or genetically modified cell lines. The cells’ response to chemicals tested are assessed with the use of a proteomics, metabolomics, genomics, or transcriptomics approach. The major tests used nowadays to evaluate the sensitizing potential in vitro are the Direct Peptide Reactivity Assay (DPRA), Human Cell Line Activation (h-CLAT), Myeloid U937 Skin Sensitization Test (MUSST), KeratinoSens method, and LuSens. 

The DPRA test is a direct determination test of peptide reactivity which relies on measuring the reactivity of the tested chemical substance with the model heptapeptides containing cysteine and lysine. The peptides are incubated with the test substance for 24 h at 25 °C. The molar ratio of cysteine to the substance tested should be 1:10 and of lysine 1:50. All samples are prepared in triplicate. Following incubation, the peptide is quantified by HPLC with UV detection at 220 nm. Cysteine and lysine peptide percentage depletion values are then calculated. To determine whether the test substance is sensitizing, the cysteine 1:10/lysine 1:50 prediction model is used (Table 2) [15,16]. 

In the h-CLAT assay, THP-1 cells are used as a surrogate for dermal dendritic cells. These cells are treated with different concentrations of test substances for 24 h, after which the expression of CD86 and CD54 markers on the cell surface is measured by flow cytometry. The relative fluorescence intensity (RFI) is used as an indicator of CD86 and CD54 expression. If the RFI of CD86 is greater than 150% or RFI of CD54 is greater than 200% of the baseline in at least two out of three experiments at any dose, then the substance is recognized as a sensitizer [16]. 

The MUSST test is based on the measurement of the CD86 marker expression in the U937 myeloid cells line. Cells are exposed for 48 h in 96-well plates to various concentrations of test substances. In the next step, the expression of the CD86 marker and the viability of the cells are evaluated by flow cytometry. If the test chemical induces an increase in CD86 expression of >70% in at least two independent experiments and the cell viability is >70%, then the substance is considered as sensitizing [16,17,18]. 

The KeratinoSens method utilizes the luciferase gene activity in modified human keratinocytes to assess cell viability. In the initial stage, the cells are incubated for 24 h, and then test substances are added at 12 different concentrations and incubated for another 48 h. After this time, the activity of luciferase and cytotoxicity (viable of cells by MTT assay) are determined. Test substances are considered sensitizing if they cause significant induction of the luciferase gene above a specific threshold in two out of three independent experiments [17,19,20]. 

LuSens is based on a principle similar to KeratinoSens but uses six different concentrations of test substances, and luminescence and cell cytotoxicity (cell viability) are measured after 48 h. The tested substance is considered as sensitizing if the luciferase activity increases by at least 1.5-fold compared to controls in at least two consecutive non-cytotoxic concentrations. An additional precondition is that at least three tested concentrations do not cause cytotoxic effects (cell viability ≥ 70%) [16,20].

Whether the described in vitro methods accurately and reliably reflect processes occurring in a much more complex system of the human body and whether they are as reliable or superior to the previously used tests in humans and animals remain a matter for debate. In vitro tests also consume relevant resources. Therefore, in the search for an alternative, we attempted to design a mathematical (in silico) model to predict the sensitizing potential of cosmetic ingredients based on historical epidemiological data. The aim of the present study was to develop an in silico model based on artificial intelligence (machine learning) to predict the sensitizing potential of haptens based on their physicochemical characteristics and to compare the predictions with historical animal data, as well as with in vitro studies to date.

## 2. Results

Having data on the prevalence of allergies in both groups—patients with ACD (Appendix A) and the general population (Appendix A)—the eOR and eAR indexes could be calculated for 50 substances (Table 3 and Table 4, respectively). In this article, under the term “substance”, we mean any substance used for patch testing, including single haptens and hapten mixes. Unique chemicals (e.g., methylparaben) or complex products retrieved naturally as such (e.g., Myroxylon pereirae resin) are referred as “haptens”. For eOR-trained classification 38 substances could be used, including 4 mixes, whereas for eAR-trained classification, 40 substances could be used, among which were 6 mixes (Table 5). These data ultimately were used as a training set for the designed in silico classification model. For each hapten, 1444 PaDEL descriptors were calculated. Sensitivity analysis resulted in 3 input vectors with a different number of descriptors: 11, 17, and 26 for eOR and 9, 17, and 24 for the eAR dataset. All the models were tested with the use of the above-described 10-fold cross-validation (CV) procedure to select the best-performing model based on its accuracy, specificity, and sensitivity. For the in silico classification system trained with eOR, the best results were obtained for a modular model (RandomComittee) that consisted of 10 decision trees trained using 17 descriptors in the input vector. For the in silico classification system trained with eAR, the best results were obtained for the model trained using 9 and 24 descriptors as the input vector. The best-performing algorithm was the Naive Bayes classifier without the kernel estimator and discretization of numerical descriptors. Due to the relatively small size of the teaching set, we decided to use a 24-element input vector in the final model to allow for a better characterization of haptens which could be included in future teaching sets to cover a more diverse chemical spectrum. The selected descriptors can be identified as a key to discrimination of molecules with regard to their sensitizing effects (Appendix A). A higher correlation was found between chemical characteristics of the molecules and sensitizing potential for the in silico eAR-trained rather than eOR-trained classification (Table 6). 

Next, we compared the results of our in silico model with published data from animal and in vitro tests (Appendix A). Out of 50 ingredients classified with eAR-trained index, the results of LLNA and in vitro tests were available for 24. Comparison between predictivity of our in silico model, LLNA and in vitro tests for the substances with available results for all three tests is presented in Table 6. For eOR-trained classification, LLNA and in vitro results were available for 22 out of 43 ingredients. Comparison between the predictivity of the in silico model, LLNA, and in vitro test for the haptens with available results for all three tests is presented in Table 7. In Table 8, substances classified incorrectly by the in silico model are listed separately for eAR-trained and eOR-trained classifications. The results of using the best-performing model (NaiveBayes, 24 parameters, accuracy 83%) to predict sensitization risk of emerging (or less studied) haptens are shown in Table 9. 

## 3. Discussion

Cosmetic science is rapidly developing; therefore, proteomics, transcriptomics, genomics, as well as machine learning have already been applied in this field, including the safety assessment of cosmetic ingredients [23,24,25,26]. In clinical research on allergic contact dermatitis, “omics” also have already been applied, e.g., to study differences between allergic and irritant reactions to chemical compounds [21,27]. Studies with such methodology grant us with an unprecedented insight into processes ongoing at the subcellular level in people who are already sensitized, yet still leave open the question of what makes a chemical more prone to sensitize people. In the present study, we attempted to develop an in silico model which would be convenient and reliable in assessing the safety of ingredients of cosmetics and topical drugs. The advantage of such infotechnomics approach is that it does not engage substantial resources such as cells, animals, or reagents and can provide results for multiple chemicals almost instantly. The proposed model predicts the sensitizing potential of a given substance based only on its structure and calculated physicochemical properties. It was trained using a learning set of ingredients classified as sensitizing or non-sensitizing based on epidemiological data retrieved from the literature and can predict the sensitizing potential of a new substance from its physicochemical and structural properties. The concordance of model predictions (in silico) with epidemiological studies in humans (real-world) is satisfactory and comparable with the current in vitro methods for assessing the sensitizing potential of topical ingredients. Bauch et al. showed that the accuracy of four different in vitro methods as compared to human patch test data ranged from 83 to 91% [22]. Urbisch et al. assessed the accuracy of in vitro test methods versus human data at 90% [16]. These results are similar to our model, which showed an accuracy of 86%. Despite the generally good performance, some substances were predicted falsely as either sensitizers or non-sensitizers (Table 8). The reason for this may be the qualitative, rather than a more subtle, quantitative classification of substances while using the eOR and eAR indexes. The eOR and eAR indexes of substances that were classified incorrectly in our in silico model were at the cut-off between sensitizing and non-sensitizing (diazolidynyl urea), human data results available for a given substance were limited (citral, ammyl cinnamyl alcohol, geraniol), or zero prevalence rates were found in epidemiological studies (isopropyl myristate, bronopol, DMDM hydantoin). Another possible source of bias is that the pooled data on the prevalence of contact sensitization among ACD patients and the general population might have originated from different countries and time periods, although we always attempted to select best fits available. Further on, some cosmetic ingredients used in routine patch testing possess certain irritant potential that might lead to false-positive reactions and overdiagnosis of allergy to a given hapten [28]. The simplistic dichotomy (sensitizer vs. non-sensitizer) should be taken with a proper dose of caution. Therefore, in our prediction model, the term “sensitizer” should be rather understood as “probably a sensitizer, more concern”, while the term “non-sensitizer” should be understood as “probably a non-sensitizer, less concern”.

This study is not without its limitations. Therefore, in order to stress that the indices used to train our AI system were computed using data collected from different populations, sometimes from different countries, we always used the prefix “estimated” while referring to estimated attributable risk (eAR) and estimated odds ratios (eOR) in our study. This possible source of bias is mainly due the scarcity of published epidemiological studies on the prevalence of contact allergy to cosmetic ingredients in the general population, and especially a lack of studies with sufficient numbers of cosmetic ingredients patch tested and medical checks carried out to single out subjects with ACD in general populations. In order to collect enough data to compute the eAR and eOR indices for a sufficient number of cosmetic haptens to train our AI model, we had to apply a relatively broad time criterion for the studies included, i.e., from 1990 onward. In order to minimize a potential bias due to differences between compared populations and changing trends in contact allergy rates, the ACD and general populations included in the calculations were best possibly matched with regard to the time of the study and geographical area, e.g., the same or neighboring countries. Ultimately, the developed AI model estimates the sensitization potential of a hapten based only on its physicochemical properties. It seems rather improbable that the interdependence between physicochemical properties of a hapten and its biological effects would noticeably change within a timeframe of less than 30 years. In addition, the main outcomes measure in epidemiological patch test studies, i.e., sensitization rates, have been measured in practically identical way within this timeframe, as the criteria for a positive patch test reaction were set by the International Contact Dermatitis Research Group in the 1970s and remain a widely accepted standard until present day. A dedicated general population study with patch testing to an extensive range of cosmetic ingredients and a parallel medical screening to single out subjects with ACD would certainly allow for computing more accurate AR and OR indices as a measure of the population effect of a hapten. An AI model trained with such unbiased measures would expectedly deliver more reliable data, allowing for more precise predictions regarding new haptens. However, we are not aware of any such epidemiological study completed to date. For the time being, it therefore seemed reasonable to attempt to make the best of the data already available. With all the caveats mentioned above, the predictions made with the present AI system trained with the available data seem rather satisfactory.

## 4. Materials and Methods

An in silico model was developed with the use of WEKA software (Waikato Environment for Knowledge Analysis, WEKA 3.8.6, The University of Waitako, Hamilton New Zeland, https://waikato.github.io/weka-wiki/downloading_weka/, accessed on 27 March 2023) [29]. WEKA is a collection of machine learning algorithms for data mining implemented in Java. WEKA offers tools for data pre-processing, classification, regression, clustering, association, and visualization. It is an Open-Source software released under the GNU Public License [30]. A database for model building consisted mainly of cosmetic ingredients classified as sensitizing or non-sensitizing based on the values of eOR (estimated odds ratio) and eAR (estimated attributable risk) indexes, which are described below. The input vector describing one substance is composed of a set of descriptors generated by the PaDEL program to characterize the molecule (PaDEL Version 2.17, Yap Chun Wei, Singapore, http://padel.nus.edu.sg, accessed on 27 March 2023) [31]. These descriptors consist of a two-dimensional structure, as well as physicochemical, topological, and electrostatic properties [31]. The output was an in silico classification of the sensitizing potential of the substance (sensitizer—“1” or non-sensitizer—“0”). A real-world classification of sensitizing potential of cosmetic ingredients based on published epidemiological data on the sensitization rates in both patients suspected of ACD (Allergic Contact Dermatitis) and the general population seemed most suitable for this study, as it reflects the health effects of haptens in a real-world situation. To date, the only validated method of the detection of contact allergy is patch testing, which is used for this purpose both in clinical and epidemiological settings [32]. Baseline series, i.e., sets of haptens most frequently used in both applications, consist of haptens that most frequently sensitize people in a given geographical area and period of time. Reflecting on the epidemiological trends, the participation of cosmetic ingredients steadily grows in the series, e.g., the present European Baseline Series includes 28 cosmetic ingredients—either separate or in mixes of 4–8 components [33]. However, cosmetics were less present in past epidemiological studies. The first stage of the present work was, therefore, to establish a list of cosmetic ingredients with sufficient epidemiological data available. For this purpose, bibliographic databases PubMed, Embase, Web of Science, and Google Scholar were searched using a query “*cosmetic ingredients*” *AND* (“*contact allergy*” *OR* “*contact hypersensitivity*” *OR* “*allergic*” *OR* “*dermatitis*” *OR* “*eczema*”). The identified components were analyzed for sensitizing potential as reported in the literature. With regard to humans, a query “*x AND patch tests*” was carried out, where “x” was the name of a cosmetic ingredient previously identified as sensitizing. The analysis also included articles cited in references of identified publications or suggested as related to the articles of interest by bibliographic databases. Data published between 1990 and 2018 (start of data collection for the purpose of this study) were included. Relevant data were extracted from the identified papers by two researchers (JK and RS) independently. The outcomes were subsequently compared and, in case of discrepancies, reanalyzed together to reach the final agreement. As the number of identified cosmetic ingredients with the necessary set of data turned out too low to feed the in silico model, data for topical drug formulations were also included in order to strengthen it. Excipients of topical drug formulations frequently share ingredients with cosmetic products, along with the formulation of the product (cream, emulsion, etc.) and method of application. For multi-ingredient mixes classified as non-sensitizers (“0”), it was assumed that each of their individual components was a non-sensitizer (“0”). Multi-component ingredients with known sensitizing potential (“1”) had to be excluded from the learning process (e.g., Fragrance Mix I, Fragrance Mix II, Methylchloroisothiazolinone/Methylisothiazolinone) as it was impossible to single out which of their ingredients would exceed the assumed thresholds due to lack of sufficient epidemiological data for individual components in ACD patients and the general population. Non-organic substances (cobalt or chromium) also had to be excluded from the database due to the limitations of the PaDEL program and an inability to provide the same set of parameters as for the organic compounds. Sorbitan sesquioleate was considered as a mix, which is in line with its actual nature. For the purpose of model development, other patch test mixes also had to be included in the learning process (Black rubber mix, Carba mix, Epoxy resin, Mercapto mix), which actually are listed as cosmetic ingredients in the Cosing database, albeit rarely used in cosmetics. Our decision was dictated by the fact that only one mix of ingredients generally associated with cosmetics (Paraben Mix) and one of the topical drugs (Quinoline mix) were available for the model, and it seemed reasonable to check how the model would handle more ingredients of mixes.

After extracting published data regarding the prevalence of contact sensitization in the group of ACD patients and in the general population, the median of reported prevalence rates was determined in both. For the subsequent analysis, eOR and eAR indexes were adopted:eOR (estimated odds ratio) is the estimated odds ratio specifying the extent to which the presence or absence of feature A (here: presence of ACD) is associated with the presence or absence of feature B (here: positive patch test) in a given population. eOR = Me_ACD_/Me_gen_Me_ACD_—median of reported sensitization rates among patients with ACD.Me_gen_—median of reported sensitization rates in the general population.eAR (estimated attributable risk) is the difference between the frequency of sensitization to a given hapten among patients with ACD and the general population, which could be interpreted as indication how many ACD cases are due to an allergy to the hapten. eAR = Me_ACD_ − Me_gen_Me_ACD_—median of the percentage of sensitization among patients with ACD.Me_gen_—median of the percentage of sensitization in the general population.

The computed eOR and eAR indicators were ordered in increasing sequence and then classified as sensitizing or non-sensitizing. Substances for which eOR was ≥3 were classified as sensitizing (coded as “1”), while remaining ones were classified as non-sensitizing (“0”). We chose an eOR threshold of 3 in analogy to the stimulation index in LLNA, where a test substance with SI ≥ 3 was considered as having a sensitizing potential [11]. For the second variant of the in silico model (eAR as endpoint), the value of 1% was adopted as the cut-off threshold: Substances for which the value of the eAR was ≥1% were classified as sensitizing (coded as “1”); the others were classified as non-sensitizing (“0”) in analogy to the criteria for including a sensitizer into baseline patch test series [34,35]. The threshold of 1% divided the analyzed substances into two fairly even subgroups, while eOR ≥ 3 produced a 1:2 division.

The PaDEL program generated 1444 descriptors characterizing the properties of a molecule. In order to limit the size of input vectors and identify key variables influencing the model performance, a sensitivity analysis was performed in WEKA software with the use of algorithms labeled as CfsSubsetEval, CorrelationAttributeEval, GainRatioAttributeEval, and InfoGainAttributeEval. The generated rankings of input variables resulted in four sets of key descriptors, including between 11 and 31 and between 9 and 37 variables for eOR and eAR, respectively. All of them were used in the model generation process. To develop a classification model for sensitizing properties, the following WEKA algorithms were tested: BayesNet, NaiveBayes, multilayer perceptrons, decision trees, and so-called expert committees in which the final decision is made by averaging the results generated by the algorithms included in the “committee”. The details of the classification models’ settings are given in Appendix A. The training sets based on above-described classification of sensitizing potential (eOR and eAR) were used to test the predictive performance of the algorithms in a 10-fold cross-validation procedure (10-CV). In this procedure, randomly selected 10% of records were excluded from the training set and from a test set. The 10 pairs of training-test sets were generated. Each algorithm (model) was then taught 10 times and tested each time using a different pair of sets. The general correctness of the prediction was assessed by comparing the predicted class with the real class determined with the eOR and eAR for each substance. The results of the binary classification were presented in the form of a matrix of errors with true positive (*TP*), true negative (*TN*), false positive (*FP*), and false negative (*FN*). Finally, we compared the predictions of our in silico model with animal and in vitro data published by expert panels [16,36,37]. On the basis of the error matrix, the classifier quality indicators were calculated according to the following formulas:

Accuracy (*AG*%)—percentage of correctly classified records:$$AG\%=\frac{TP+TN}{TP+FP+TN+FN}*100\%$$

Sensitivity (*SE*):$$SE=\frac{TP}{TP+FN}$$

Specificity (*SP*):$$SP=\frac{TN}{TN+FP}$$

The last step was to test the applicability of the trained in silico model. For this, we used the model with best accuracy (NaiveBayes eAR, 24 parameters) to assess the sensitizing potential of 15 selected haptens based only on their physicochemical and structural characteristics. The haptens were arbitrarily selected for analysis based on two criteria: (1) they were of rising interest to the scientific community as possible new sensitizers or were of special interest to our group in connection to an ongoing study, and (2) their sensitizing potential was still subject to debate at the time of selection. The resulting classification is therefore a kind of a forecast to be verified by future epidemiological data. The model structure and training datasets for the two best performing eAR and eOR-based models developed in the present research can be found in Appendix A to this article: 24eAR_NB.model—eAR index Naive Bayes model with 24 input variables, eAR_24in.arff—learning set for eAR index model, 17eOR_RC.model—eOR index Random Comittee model with 17 input variables, eOR_17in.arff—learning set for eOR index model. These files can be opened and tested in the WEKA software. Assessment of a new compound requires a generation of a test set where the compound is described with the defined set of descriptors.

## 5. Conclusions

The in silico approach shown in this paper allows for a prediction of the sensitizing potential of haptens with accuracy comparable to the historical animal tests and in vitro tests used nowadays. In silico models consume little resources, are free of ethical concerns, and can provide results for multiple chemicals almost instantly; therefore, the proposed infotechnomics approach seems useful in the safety assessment of cosmetics.

## Figures and Tables

**Table 1 ijms-24-06801-t001:** A timetable of the withdrawal of animal tests for cosmetic ingredients in the European Union.

	Before 2004	2004	2009	2013
Testing of finished cosmetic products on animals, where alternative methods are available	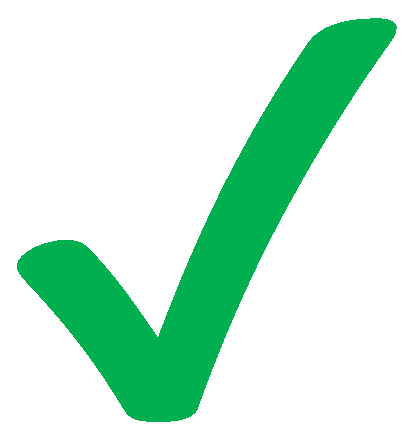	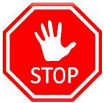	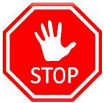	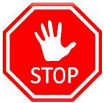
Testing of finished cosmetic products on animals, if there are no alternative methods	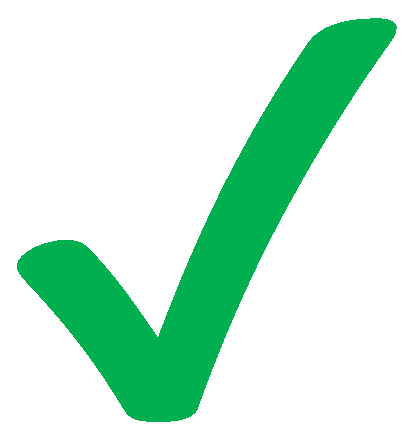	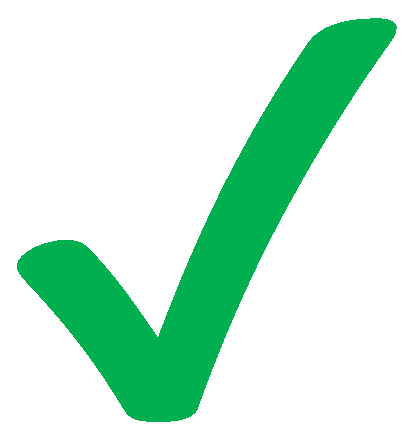	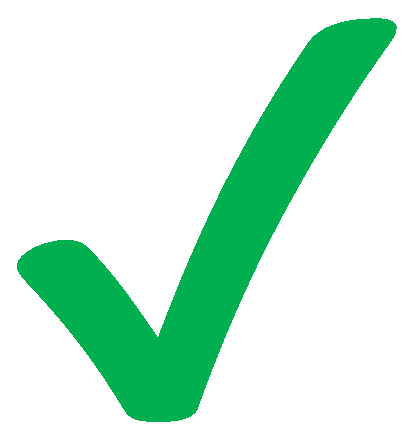	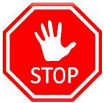
Testing of cosmetic ingredients on animals, where alternative methods are available	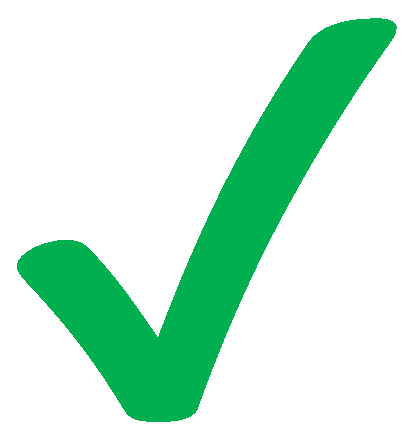	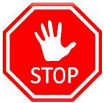	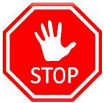	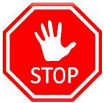
Testing of cosmetics ingredients on animals, if there are no alternative methods	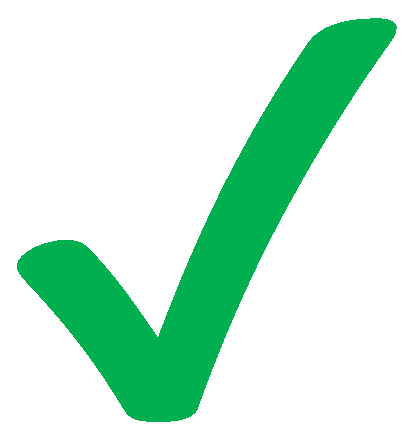	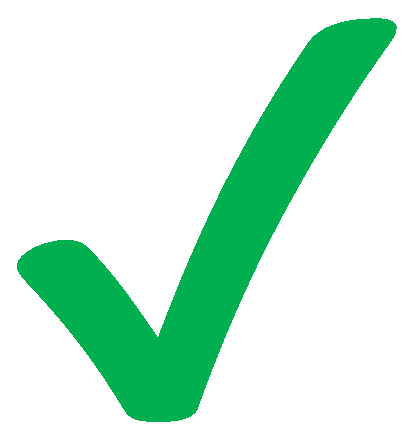	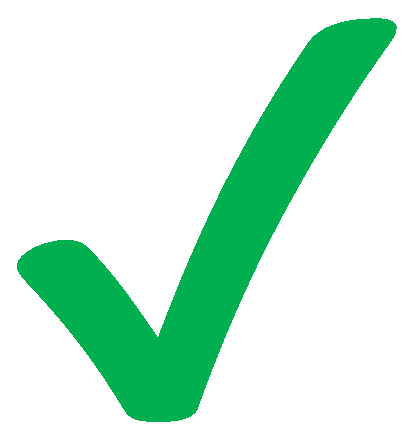	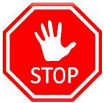
Sale of cosmetic products tested on animals, where alternative methods are available	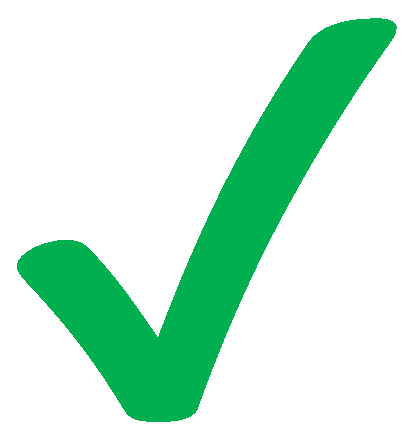	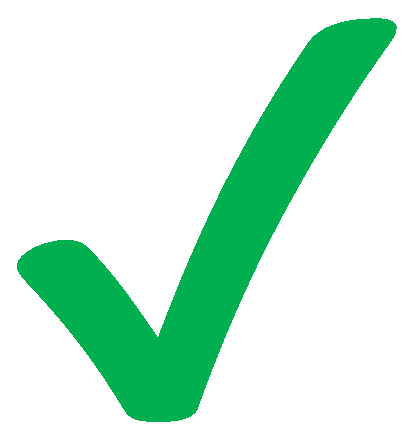	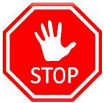	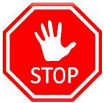
Sale of cosmetic products tested on animals for repeated dose toxicity, reproductive toxicity and toxicokinetics, for which there are no alternative methods yet	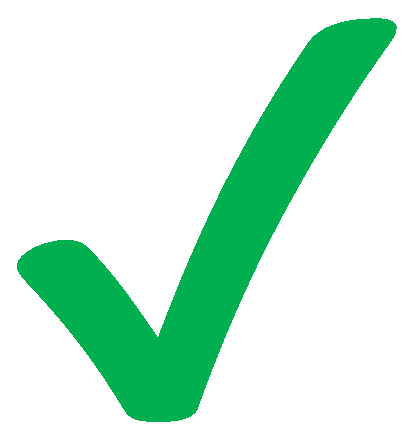	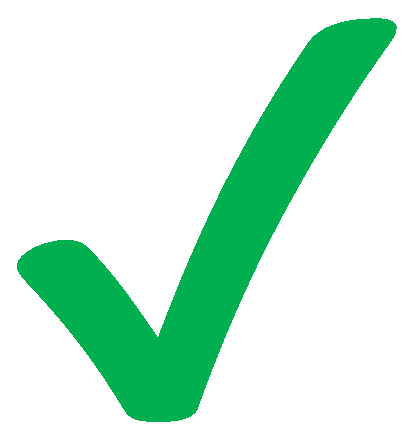	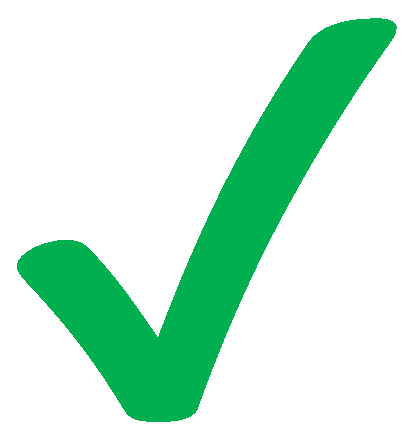	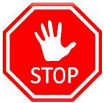
Placing on the market substances tested on animals, if there are no alternative methods	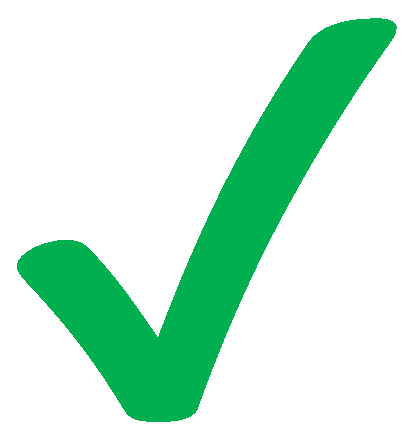	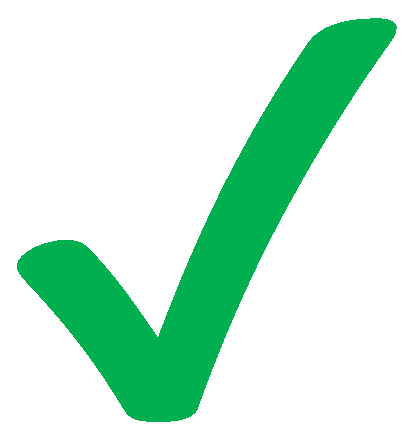	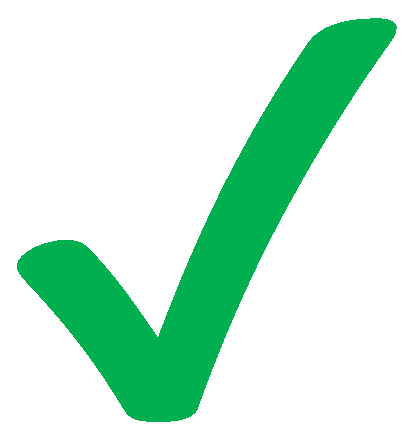	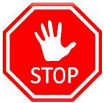

**Table 2 ijms-24-06801-t002:** Cysteine 1:10/lysine 1:50 prediction model according to OECD TG 442 C 2015.

Mean of Cysteine and Lysine % Depletion ^†^	Reactivity Class	Prediction
0% < mean % depletion < 6.38%	Minimal reactivity	Non-sensitizer
6.38% < mean % depletion < 22.62%	Low reactivity	Sensitizer
22.62% < mean % depletion < 42.47%	Moderate reactivity	Sensitizer
42.47% < mean % depletion < 100%	High reactivity	Sensitizer

^†^ Ranges according to original publication—in fact, the original open ranges offer no interpretation for the values 6.38%, 22.62%, 42.47%, and 100%.

**Table 3 ijms-24-06801-t003:** Calculation of the estimated odds ratio (eOR) for the analyzed substances.

No.	Substance (Hapten or Hapten Mixes)	Me_ACD_ [%]	Me_gen_ [%]	eOR	0/1 Classification ^†^	In Silico Utilization
1.	Benzocaine	0.50	1.00	0.50	0	included
2.	Hydroxyisohexyl 3-cyclohexene carboxaldehyde (HICC)—Lyral	2.35	2.00	1.17	0	included
3.	Sorbitan sesquioleate	0.60	0.50	1.20	0	included
4.	Iodopropynyl butylcarbamate	0.85	0.60	1.42	0	included
5.	Benzyl salicylate	0.75	0.50	1.50	0	included
6.	Cocamidopropyl betaine	3.20	2.00	1.60	0	included
7.	Cobalt (di)chloride	4.90	2.80	1.75	0	excluded
8.	Methyldibromo Glutaronitrile (MDBGN)	2.80	1.60	1.75	0	included
9.	Lanolin	1.70	0.95	1.79	0	included
10.	Evernia prunastri extract	1.55	0.75	2.06	0	included
11.	Epoxy resin	1.25	0.60	2.08	0	included
12.	Imidazolidynyl urea	1.30	0.50	2.60	0	included
13.	Farnesol	1.05	0.40	2.62	0	included
14.	Mercapto mix	0.80	0.30	2.67	0	included
15.	Methylisothiazolinone (MI)	3.90	1.45	2.69	0	included
16.	Quinoline mix	1.10	0.40	2.75	0	included
17.	Diazolidynyl urea	1.50	0.50	3.00	1	included
18.	Ammylcinnamyl alcohol	0.30	0.10	3.00	1	included
19.	Mercaptobenzothiazole	0.60	0.20	3.00	1	included
20.	Thimerosal	10.20	3.40	3.00	1	included
21.	Formaldehyde	2.60	0.80	3.25	1	included
22.	Paraben mix	1.00	0.30	3.33	1	excluded
23.	Colophonium	3.90	1.10	3.55	1	included
24.	Fragrance Mix I (FM I)	9.40	2.45	3.87	1	excluded
25.	Black rubber mix	0.85	0.20	4.25	1	excluded
26.	Quaternium 15	1.85	0.40	4.63	1	included
27.	p-Phenylenediamine (PPD)	4.80	1.00	4.80	1	included
28.	Thiuram mix	2.70	0.50	5.40	1	excluded
29.	Fragrance Mix II (FM II)	4.90	0.90	5.44	1	excluded
30.	Methylchloroisothiazolinone/Methylisothiazolinone (MCI/MI)	2.80	0.50	5.60	1	excluded
31.	Myroxylon pereirae resin	6.50	1.10	5.90	1	included
32.	Citral	1.20	0.20	6.00	1	included
33.	Cinnamal	5.15	0.80	6.44	1	included
34.	Potassium dichromate	5.40	0.80	6.75	1	excluded
35.	Hydroxycitronellal	4.05	0.50	8.10	1	included
36.	Carba mix	4.60	0.50	9.20	1	excluded
37.	Ethylenediamine (dichloride)	2.40	0.25	9.60	1	excluded
38.	Geraniol	4.25	0.40	10.62	1	included
39.	Wool alcohols	2.40	0.20	12.00	1	excluded
40.	Neomycin sulfate	5.05	0.40	12.63	1	included
41.	Caine mix	1.30	0.10	13.00	1	excluded
42.	Methylchloroisothiazolinone (MCI)	4.00	0.20	20.00	1	included
43.	Cinnamic alcohol	8.30	0.30	27.77	1	included
44.	Isopropyl myristate	0.20	0.00	NC	1	included
45.	Benzyl alcohol	0.30	0.00	NC	1	included
46.	Propyl gallate	0.70	0.00	NC	1	included
47.	Triethanolamine	0.80	0.00	NC	1	included
48.	Bronopol	1.20	0.00	NC	1	included
49.	DMDM Hydantoin	1.35	0.00	NC	1	included
50.	Butylhydroxyanisole (BHA)	1.40	0.00	NC	1	included

^†^ Substances for which the value of the eOR index was ≥3 were classified as sensitizing (coded as “1”); NC—not calculable (division by 0). Me_ACD_—median of reported sensitization rates among patients with ACD; Me_gen_—median of reported sensitization rates in the general population.

**Table 4 ijms-24-06801-t004:** Calculation of the estimated attributable risk (eAR) for the analyzed substances.

No.	Substance (Hapten or Hapten Mixes)	Me_ACD_ [%]	Me_gen_ [%]	eAR [%]	0/1 Classification ^†^	In Silico Utilization
1.	Benzocaine	0.50	1.00	−0.50	0	included
2.	Sorbitan sesquioleate	0.60	0.50	0.10	0	included
3.	Ammylcinnamyl alcohol	0.30	0.10	0.20	0	included
4.	Isopropyl myristate	0.20	0.00	0.20	0	included
5.	Benzyl salicylate	0.75	0.50	0.25	0	included
6.	Iodopropynyl butylcarbamate	0.85	0.30	0.25	0	included
7.	Benzyl alcohol	0.30	0.00	0.30	0	included
8.	Hydroxyisohexyl 3-cyclohexene carboxaldehyde (HICC, Lyral)	2.35	2.00	0.35	0	included
9.	Citral	0.60	0.20	0.40	0	included
10.	Mercaptobenzothiazole	0.60	0.20	0.40	0	included
11.	Mercapto mix	0.80	0.30	0.50	0	included
12.	Farnesol	1.05	0.40	0.65	0	included
13.	Black rubber mix	0.85	0.20	0.65	0	included
14.	Epoxy resin	1.25	0.60	0.65	0	included
15.	Paraben mix	1.00	0.30	0.70	0	included
16.	Propyl gallate	0.70	0.00	0.70	0	included
17.	Quinoline mix	1.10	0.40	0.70	0	included
18.	Evernia prunastri extract	1.55	0.75	0.75	0	included
19.	Lanolin	1.70	0.95	0.75	0	included
20.	Imidazolidynyl urea	1.30	0.50	0.80	0	included
21.	Triethanolamine	0.80	0.00	0.80	0	included
22.	Diazolidynyl urea	1.50	0.50	1.00	1	included
23.	Bronopol	1.20	0.00	1.20	1	included
24.	Cocamidopropyl betaine	3.20	2.00	1.20	1	included
25.	Methyldibromo Glutaronitrile (MDBGN)	2.80	1.60	1.20	1	included
26.	Caine mix	1.30	0.10	1.20	1	excluded
27.	DMDM Hydantoin	1.35	0.00	1.35	1	included
28.	Butylhydroxyanisole (BHA)	1.40	0.00	1.40	1	included
29.	Quaternium 15	1.85	0.40	1.45	1	included
30.	Formaldehyde	2.60	0.80	1.80	1	included
31.	Cobalt (di)chloride	4.90	2.80	2.10	1	excluded
32.	Ethylenediamine (dichloride)	2.40	0.25	2.15	1	excluded
33.	Colophonium	3.40	1.20	2.20	1	included
34.	Wool alcohols	2.40	0.20	2.20	1	excluded
35.	Thiuram mix	2.70	0.50	2.20	1	excluded
36.	Methylchloroisothiazolinone/Methylisothiazolinone (MCI/MI)	2.80	0.50	2.30	1	excluded
37.	Methylisothiazolinone (MI)	3.90	1.45	2.45	1	included
38.	Hydroxycitronellal	4.05	0.50	3.55	1	included
39.	Methylchloroisothiazolinone (MCI)	4.00	0.20	3.80	1	included
40.	p-Phenylenediamine (PPD)	4.80	1.00	3.80	1	included
41.	Geraniol	4.25	0.40	3.85	1	included
42.	Fragrance Mix II (FM II)	4.90	0.90	4.00	1	excluded
43.	Carba mix	4.60	0.50	4.10	1	excluded
44.	Cinnamal	5.15	0.80	4.35	1	included
45.	Potassium dichromate	5.40	0.80	4.60	1	excluded
46.	Neomycin sulfate	5.05	0.40	4.65	1	included
47.	Myroxylon pereirae resin	6.50	1.10	5.40	1	included
48.	Thimerosal	10.20	3.40	6.80	1	included
49.	Fragrance Mix I (FM I)	9.40	2.45	6.95	1	excluded
50.	Cinnamic alcohol	8.30	0.30	8.00	1	included

^†^ Substances for which the value of the eAR index was ≥1% were classified as sensitizing (coded as “1”); the remaining ones were considered as non-sensitizers (“0”). Me_ACD_—median of the percentage of sensitization among patients with ACD; Me_gen_—median of the percentage of sensitization in the general population.

**Table 5 ijms-24-06801-t005:** Substances and haptens qualified for the eAR- and eOR-trained model database.

No.	Substance	Hapten	eAR	eOR
1.	Ammylcinnamyl alcohol	Ammylcinnamyl alcohol	0	1
2.	Benzocaine	Benzocaine	0	0
3.	Benzyl alcohol	Benzyl alcohol	0	1
4.	Benzyl salicylate	Benzyl salicylate	0	0
5.	Black rubber mix	N-isopropyl-N-phenyl parapheylenediamine	0	**1** ^†^
		N-cyclohexyl-N-phenyl paraphenylenediamine	0	
		N-biphenyl paraphenylenediamine	0	
6.	Butylhydroxyanisole (BHA)	Butylhydroxyanisole	1	1
7.	Bronopol	Bronopol	1	1
8.	Cinnamal	Cinnamal	1	1
9.	Cinnamic alcohol	Cinnamic alcohol	1	1
10.	Citral	Citral	0	1
11.	Cocamidopropyl betaine	Cocamidopropyl betaine	1	0
12.	Colophonium	Colophonium	1	1
13.	Diazolidynyl urea	Diazolidynyl Urea	1	1
14.	DMDM Hydantoin	DMDM Hydantoin	1	1
15.	Epoxy resin	Epichlorohydrin	0	0
		4,4′-Isopropylidenediphenol	0	0
16.	Evernia prunastri extract	Evernia prunastri extract	0	0
17.	Farnesol	Farnesol	0	0
18.	Formaldehyde	Formaldehyde	1	1
19.	Geraniol	Geraniol	1	1
20.	Hydroxycitronellal	Hydroxycitronellal	1	1
21.	Imidazolidynyl urea	Imidazolidynyl urea	0	0
22.	Iodopropynyl butylcarbamate	Iodopropynyl butylcarbamate	0	0
23.	Isopropyl myristate	Isopropyl myristate	0	1
24.	Lanolin	Lanolin	0	0
25.	Hydroxyisohexyl 3-cyclohexene carboxaldehyde (HICC, Lyral)	Hydroxyisohexyl 3-cyclohexene carboxaldehyde	0	0
26.	Mercapto mix	2,’2-Benzothiazyl_disulfide	0	0
		4-Morpholinyl-2-benzothiazyl disulfide	0	0
		3_N-Cyclohexyl-2-benzothiazolesulfenamide	0	0
27.	Mercaptobenzothiazole	Mercaptobenzothiazole	0	1
28.	Methylchloroisothiazolinone (MCI)	Methylchloroisothiazolinone	1	1
29.	Methyldibromo Glutaronitrile (MDBGN)	Methyldibromo glutaronitrile	1	0
30.	Methylisothiazolinone (MI)	Methylisothiazolinone	1	0
31.	Myroxylon pereirae resin	Myroxylon pereirae resin	1	1
32.	Neomycin sulfate	Neomycin sulfate	1	1
33.	Paraben mix	Methylparaben	0	**1** ^†^
		Bythylparaben	0	
		Ethylparaben	0	
		Propylparaben	0	
34.	p-Phenylenediamine (PPD)	p-Phenylenediamine	1	1
35.	Propyl gallate	Propyl gallate	0	1
36.	Quaternium 15	Quaternium 15	1	1
37.	Quinoline mix	Quinoline	0	0
		Chlorquinadol	0	0
38.	Sorbitan sesquilate	Sorbitol	0	0
		Oleic acid	0	0
39.	Thimerosal	Thimerosal	1	1
40.	Triethanolamine	Triethanolamine	0	1
Total numer of haptens usable in the in silico model			50	43

^†^ haptens excluded from in silico model, see explanations in the text.

**Table 6 ijms-24-06801-t006:** Predictivity of our in silico model based on eAR classification versus published data on LLNA and in vitro tests.

eAR *n* = 24	In Silico (NaiveBayes)	LLNA [15,21,22]	In Vitro [15,21,22]
Accuracy	0.83	0.58	0.50
Sensitivity	0.82	0.91	0.82
Specificity	0.85	0.31	0.23
False Positive	0.15	0.20	0.40
False Negative	0.18	0.47	0.53

**Table 7 ijms-24-06801-t007:** Predictivity of our in silico model based on eOR classification versus published data on LLNA and in vitro tests.

eOR *n* = 22	In Silico (RandomComitee)	LLNA [15,21,22]	In Vitro [15,21,22]
Accuracy	0.73	0.68	0.59
Sensitivity	0.71	0.93	0.79
Specificity	0.75	0.25	0.25
False Positive	0.40	0.33	0.60
False Negative	0.17	0.32	0.35

**Table 8 ijms-24-06801-t008:** Substances classified incorrectly by the in silico model.

eAR			eOR		
Hapten	Real Class ^†^	Predicted Class ^‡^	Haptens	Real Class ^†^	Predicted Class ^‡^
Ammylcinnamyl alcohol	0	1	Benzocaine	0	1
Citral	0	1	Benzyl salicylate	0	1
Geraniol	1	0	Bronopol	1	0
Hydroxyisohexyl 3-cyclohexene carboxaldehyde (HICC, Lyral)	0	1	Diazolidynyl urea	1	0
Neomycin	1	0	DMDM Hydantoin	1	0
p-Phenylenediamine	1	0	Hydroxycitronellal	1	0
Thimerosal	1	0	Isopropyl myristate	1	0
			Methylchloroisothiazolinone	1	0
			Myroxylon pereirae resin	1	0
			Sorbitol	0	1

^†^ Real class—classification based on, respectively, eAR or eOR indexes calculated from published epidemiological data. ^‡^ Predicted class—classification based on in silico model (0—non-sensitizer; 1—sensitizer).

**Table 9 ijms-24-06801-t009:** Predictions on the sensitization risk of selected haptens using in silico model with the highest accuracy (NaiveBayes eAR, 24 parameters).

Hapten	IUPAC Name	CAS RN	Predicted
Acetophenone azine	(E)-1-phenyl-N-[(E)-1-phenylethylideneamino]ethanimine	729-43-1	0
Cyclamen aldehyde	2-methyl-3-(4-propan-2-ylphenyl)propanal	103-95-7	1
Dibucaine	2-butoxy-N-[2-(diethylamino)ethyl]quinoline-4-carboxamide	85-79-0	0
N,N-Dimethylacrylamide	N,N-dimethylprop-2-enamide	2680-03-7	1
Dimethylthiocarbamyl benzothiazole sulphide	N,N-dimethylthiocarbamylbenzothiazole sulfide	3432-25-5	1
Disperse Blue 106	2-(Ethyl(3-methyl-4-((5-nitrothiazol-2-yl)diazenyl)phenyl)amino)ethanol	68516-81-4	0
Disperse Blue 124	2-[N-ethyl-3-methyl-4-[(5-nitro-1,3-thiazol-2-yl)diazenyl]anilino]ethyl acetate	15141-18-1	0
Geraniol hydroperoxide	(2Z)-1-hydroperoxy-3,7-dimethylocta-2,6-dien-1-ol	n/a	1
Hexyl salicylate	hexyl 2-hydroxybenzoate	6259-76-3	0
Isobornyl acrylate	[(1R,2R,4R)-1,7,7-trimethyl-2-bicyclo[2.2.1]heptanyl] prop-2-enoate	5888-33-5	1
Lidocaine	2-(diethylamino)-N-(2,6-dimethylphenyl)acetamide	137-58-6	0
Neral	(2Z)-3,7-dimethylocta-2,6-dienal	106-26-3	1
Prenyl caffeate	3-methylbut-2-enyl (E)-3-(3,4-dihydroxyphenyl)prop-2-enoate	118971-61-2	1
Tetracaine	2-(dimethylamino)ethyl 4-(butylamino)benzoate	94-24-6	0
2-(thiocyanomethylthio) benzothiazole	1,3-benzothiazol-2-ylsulfanylmethyl thiocyanate	21564-17-0	0

Predicted: 0—“non-sensitizer”, 1—“sensitizer”.

## Data Availability

The data used in this study are available in Appendix A.

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
