# Peer review of "Artificial Intelligence That Predicts Sensitizing Potential of Cosmetic Ingredients with Accuracy Comparable to Animal and In Vitro Tests—How Does the Infotechnomics Compare to Other “Omics” in the Cosmetics Safety Assessment?"

_ijms, 2023, doi:10.3390/ijms24076801_

Round 1
Reviewer 1 Report
The authors address an interesting topic in cosmetics: testing sensitizing potential free of ethical concerns. However, some points should be more elucidated:
- The references published after 2016 are practically missing.
- This also concerns the very beginning of the Introduction, where the first paragraph justifying the topic of the manuscript is based on the literature published before 20, 30 years and the question is whether the conclusions (also numerical) from that period can be projected to present time.
- As this manuscript is not intended as a revision, a series of papers do not match the presented topic (and are out of date).
- Reaction to cosmetic products also depends on geographical area and seasons. Are there any differences in modelling?
- Is your model available at disposal and if yes, where? If not then the impact of your presentation is rather reduced.
- The section Conclusion does not conclude your investigation and is based only on very general sentences not contributing to understanding your findings.
Author Response
Response to Reviewer 1 Comments
Reviewer:
- The references published after 2016 are practically missing.
Reply: Thank you for the above comments. The literature sources have been updated wherever possible. So now the number of recent papers has substantially increased wherever possible. Nevertheless, keeping some older references on the list seemed necessary for the following reasons:
1) The Method of this study was based on all available human epidemiological data published after 1990, which indeed is 3 decades. This timespan resulted mainly from the scarcity of data regarding the prevalence of contact allergy to cosmetic ingredients from the general population. So while calculating eARs and eORs used to teach the AI, we had to build around these scarce general population data and pair them with studies of ACD patients fitting best with regard to geographical area and period in time.
The compared groups (ACD patients versus general population) selected for the eAR and eOR indices were matched with regard to geographical area and time period in order to minimize the possible bias due to time trends and geographical differences in sensitisation rates. The developed AI model assesses the sensitization potential based only on physicochemical properties of the haptens, and it seems improbable that the interdependence between properties of a molecule vs its biological effects in human or animal body could change within 30 years. In another words, it seems rather improbable that relationships between molecular properties and sensitizing potential would significantly differ between 1990 and present days because neither physical properties of the molecules studied, nor the physiological mechanisms in humans or animals have changed significantly over the 30 years. The understanding of physiological processes might well have improved in the meantime, however, human or animal organisms are treated in the model a "black box" with the same outcomes measured today as 30 years ago. As for the training phase of the system, the reading and grading of patch test reaction have not significanly changed within the covered period - it is still the ICDRG grading of the 1970s.
2) the Introduction contains a review of historical animal tests used in cosmetic safety assessment. The sources detailing the methods are indeed not new which seem rather unavoidable taking the circumstances. However, we believe that a review of the historical methods provides added value to the paper, which will be discussed in more detail below.
Reviewer:
- This also concerns the very beginning of the Introduction, where the first paragraph justifying the topic of the manuscript is based on the literature published before 20, 30 years and the question is whether the conclusions (also numerical) from that period can be projected to present time.
Reply: In line with the Reviewer's suggestions, the introductory paragraph was rewritten based on newest literature relevant to the topic.
Reviewer:
- As this manuscript is not intended as a revision, a series of papers do not match the presented topic (and are out of date).
Reply: It is our understanding that this remark refers to the cited papers on historical animal tests. The rationale of including a "mini-review" of historical animal methods into the Introduction is as follows: for a long time, animal tests were considered as "gold standard" in testing cosmetic ingredients. After the EU ban on animal tests (as outlined in Table 1), the new in vitro methods have been developed to replace them, however, it is still a work in progress. Our AI model is an alternative attempt at the safety assessment, where an AI tool is trained on available human epidemiological data and later its outcomes compared both with historical animal data and present in vitro data. As not all readers are familiar with the banned animal tests and the present in vitro tests, we do believe that a concise overview of the methods that we propose an alternative for is helpful to the reader as we use the data obtained with these methods comparators with our AI system.
Alternatively, we could remove the entire overview of the animal and in vitro methods. In our opinion, this is a valuable background information that helps the readers to understand better the context of this study, but we leave this decision at the discretion of the Editors.
Reviewer:
- Reaction to cosmetic products also depends on geographical area and seasons. Are there any differences in modelling?
Reply: The present study was aimed at assessing the sensitizing potential of a hapten based on its physicochemical properties, rather than to analyse epidemiological trends including seasonal variations. The input data (ACD patients vs general populations) used for the AI training were matched with regard to geographical regions and period in time. However, patch test studies were typically done over a longer period of time (several years) and data shown collectively regardless the season they were collected in. Therefore, seasonal differences could not be extracted from the original studies used to train the AI system.
Reviewer:
- Is your model available at disposal and if yes, where? If not then the impact of your presentation is rather reduced.
Reply: In line with the Reviewer's suggestion, the WEKA files of our model are now available as the supplementary material to our paper.
Reviewer:
- The section Conclusion does not conclude your investigation and is based only on very general sentences not contributing to understanding your findings.
Reply: The Conclusions were rewritten and made more specific.
Reviewer 2 Report
Abstract • Please provide further information about the study outcomes • In the abstract you need to answer the following questions, what, why and how and discuss the study new findings, limitations, and future research Introduction - discuss the research aims, research gap and discuss the paper layout Add up-to-date references to support your discussion - The necessity and innovation of the article should be presented to the introduction - The literature reviewed and cited is in the main rather old. There are about many recent researches published on this topic, please cite the following articles: 1. Jairoun AA, Al-Hemyari SS, Shahwan M, El-Dahiyat F, Zyoud SE, Jairoun O, Shayeb MA. Development and Validation of an Instrument to Appraise the Tolerability, Safety of Use, and Pleasantness of a Cosmetic Product. Cosmetics. 2023 Jan 12;10(1):15. 2. Jairoun, A.A.; Al-Hemyari, S.S.; Shahwan, M.; Zyoud, S.H. An Investigation into Incidences of Microbial Contamination in Cosmeceuticals in the UAE: Imbalances between Preservation and Microbial Contamination. Cosmetics 2020, 7, 92. https://doi.org/10.3390/cosmetics7040092 6. 3. Jairoun AA, Al-Hemyari SS, Shahwan M, Zyoud SH, Ashames A. Hidden Formaldehyde Content in Cosmeceuticals Containing Preservatives that Release Formaldehyde and Their Compliance Behaviors: Bridging the Gap between Compliance and Local Regulation. Cosmetics. 2020; 7(4):93. https://doi.org/10.3390/cosmetics7040093 7. Methods • The methodology of this study should be detailed, limit information was provided on method and materials. • Validation of the study questionnaire is missing in this study which affect the internal and external validity. Discussion - I believe that more in depth discussion is needed. The discussion as present now is simple and concise. Revision of more papers using similar technique is needed - In the discussion, please discuss if the study research questions are answered or not Also introduce the model in detail. Draw a conclusion from this study and present the limitations and future research.
Author Response
Response to Reviewer 2 Comments
Reviewer:
Abstract • Please provide further information about the study outcomes • In the abstract you need to answer the following questions, what, why and how and discuss the study new findings, limitations, and future research
Reply: Thank you for the suggestions. We have added more information about the study outcomes in the Abstract while trying to stay within resonable word count limits.
Reviewer:
Introduction - discuss the research aims, research gap and discuss the paper layout Add up-to-date references to support your discussion - The necessity and innovation of the article should be presented to the introduction - The literature reviewed and cited is in the main rather old.
Reply: Thank you, below ist the inventory of fragments in the text that address the key points:
#research aims: "The aim of the present study was to develop an in silico model based on artificial intelligence (machine learning) to predict the sensitizing potential of haptens based on their physicochemical characteristics and to compare the predictions with historical animal data, as well as with in vitro studies of nowadays."
#research gap: "Whether the described in vitro methods accurately and reliably re-flect processes occurring in a much more complex system of the human body and whether they are as reliable or superior to the previously used tests in humans and animals remains a matter for debate. "
#necessity and innovation of the article: as above, plus the following sentence "In vitro tests also consume relevant resources. Therefore, in the search for an alternative, we attempted at designing a mathematical (in silico) model to predict sensitizing potential of cosmetic ingredients based on historical epidemiological data."
#discuss the paper layout: we have the impression that discussing specifically a layout of an academic text is practised in academic theses and essays, rather than journal articles which have a rather unified structure (introduction, aim, matherial and methods, results, discussions, conclusions, references) and that the layout of our article is quite self-explanatory.
Reviewer:
Add up-to-date references;
literature reviewed and cited is in the main rather old
Reply: Thank you for the above comments. The literature sources have been updated wherever possible. So now the number of recent papers has substantially increased wherever possible. Nevertheless, keeping some older references on the list seemed necessary for the following reasons:
1) The Method of this study was based on all available human epidemiological data published after 1990, which indeed is 3 decades. This timespan resulted mainly from the scarcity of data regarding the prevalence of contact allergy to cosmetic ingredients from the general population. So while calculating eARs and eORs used to teach the AI, we had to build around these scarce general population data and pair them with studies of ACD patients fitting best with regard to geographical area and period in time.
The compared groups (ACD patients versus general population) selected for the eAR and eOR indices were matched with regard to geographical area and time period in order to minimize the possible bias due to time trends and geographical differences in sensitisation rates. The developed AI model assesses the sensitization potential based only on physicochemical properties of the haptens, and it seems improbable that the interdependence between properties of a molecule vs its biological effects in human or animal body could change within 30 years. In another words, it seems rather improbable that relationships between molecular properties and sensitizing potential would significantly differ between 1990 and present days because neither physical properties of the molecules studied, nor the physiological mechanisms in humans or animals have changed significantly over the 30 years. The understanding of physiological processes might well have improved in the meantime, however, human or animal organisms are treated in the model a "black box" with the same outcomes measured today as 30 years ago. As for the training phase of the system, the reading and grading of patch test reaction have not significanly changed within the covered period - it is still the ICDRG grading of the 1970s.
2) the Introduction contains a review of historical animal tests used in cosmetic safety assessment. The sources detailing the methods are indeed not new which seem rather unavoidable taking the circumstances. However, we believe that a review of the historical methods provides added value to the paper.
Reviewer:
There are about many recent researches published on this topic, please cite the following articles: 1. Jairoun AA, Al-Hemyari SS, Shahwan M, El-Dahiyat F, Zyoud SE, Jairoun O, Shayeb MA. Development and Validation of an Instrument to Appraise the Tolerability, Safety of Use, and Pleasantness of a Cosmetic Product. Cosmetics. 2023 Jan 12;10(1):15. 2. Jairoun, A.A.; Al-Hemyari, S.S.; Shahwan, M.; Zyoud, S.H. An Investigation into Incidences of Microbial Contamination in Cosmeceuticals in the UAE: Imbalances between Preservation and Microbial Contamination. Cosmetics 2020, 7, 92. https://doi.org/10.3390/cosmetics7040092 6. 3. Jairoun AA, Al-Hemyari SS, Shahwan M, Zyoud SH, Ashames A. Hidden Formaldehyde Content in Cosmeceuticals Containing Preservatives that Release Formaldehyde and Their Compliance Behaviors: Bridging the Gap between Compliance and Local Regulation. Cosmetics. 2020; 7(4):93. https://doi.org/10.3390/cosmetics7040093 7.
Methods • The methodology of this study should be detailed, limit information was provided on method and materials. • Validation of the study questionnaire is missing in this study which affect the internal and external validity.
Reply: Thank you for the suggestion, the above papers have been incorporated into our paper and added to the reference list.
Reviewer:
Discussion - I believe that more in depth discussion is needed. The discussion as present now is simple and concise. Revision of more papers using similar technique is needed - In the discussion, please discuss if the study research questions are answered or not Also introduce the model in detail. Draw a conclusion from this study and present the limitations and future research.
Reply: The discussion has been expanded and an paragraph on study limitations was added in Discussion. The description of our model was expanded and files allowing other scientists to test our model using the WEKA software were added as supplementary material.
Round 2
Reviewer 1 Report
The authors addressed all the comments raised in my review.